# Role of Leaf Litter in Above-Ground Wood Decay

**DOI:** 10.3390/microorganisms8050696

**Published:** 2020-05-09

**Authors:** Grant T. Kirker, Amy Bishell, Jed Cappellazzi, Jonathan Palmer, Nathan Bechle, Patricia Lebow, Stan Lebow

**Affiliations:** 1USDA-FS Forest Products Laboratory, Madison, WI 53726, USA; amy.b.bishell@usda.gov (A.B.); nathan.bechle@usda.gov (N.B.); patricia.k.lebow@usda.gov (P.L.); stan.lebow@usda.gov (S.L.); 2Dept. of Wood Science and Engineering, Oregon State University, Corvallis, OR 97731, USA; Jed.Cappellazzi@oregonstate.edu; 3Northern Research Station, USDA Forest Service, Madison, WI 53726, USA; palmer.jona@gmail.com

**Keywords:** wood decay fungi, saprophytic fungi, above-ground exposure, leaf litter, moisture content, amplicon-based sequencing

## Abstract

The effects of leaf litter on moisture content and fungal decay development in above-ground wood specimens were assessed. Untreated southern pine specimens were exposed with or without leaf litter contact. Two types of leaf litter were evaluated; aged (decomposed) and young (early stages of decomposition). The moisture content of specimens was monitored, and specimens were periodically removed for visual evaluation of decay development. In addition, amplicon-based sequencing analysis of specimens and associated leaf litter was conducted at two time points. Contact with either type of leaf litter resulted in consistently higher moisture contents than those not in contact with leaf litter. Visually, evident decay developed most rapidly in specimens in contact with the aged leaf litter. Analysis of amplicon-based sequencing revealed that leaf litter contributes a significant amount of the available wood decay fungal community with similar communities found in the litter exposed wood and litter itself, but dissimilar community profiles from unexposed wood. Dominant species and guild composition shifted over time, beginning initially with more leaf saprophytes (ascomycetes) and over time shifting to more wood rotting fungi (basidiomycetes). These results highlight the importance of the contributions of leaf litter to fungal colonization and subsequent decay hazard for above-ground wood.

## 1. Introduction

Soil contact presents a severe deterioration hazard for wood products, but the greatest volume of wood products used outdoors is not in direct contact with the ground. In the last decade, there has been increasing interest in using less toxic preservative systems or lower preservative retentions for wood used above-ground. These preservative formulations may not be evaluated with ground-contact stake tests, and instead are evaluated using above-ground test methods. There are several standardized above-ground test methods [1], but accelerated evaluation of wood products intended for use above-ground has proven more difficult than ground contact evaluations. It remains unclear how well above-ground tests characterize the hazard, or if they actually accelerate the rate of decay relative to in-service applications. Most methods utilize some type of joint, connection, or layering in an effort to trap moisture, but this effect can be undermined using specimens with small dimensions. Although the smaller dimensions do allow more rapid detection of decay once it is present, smaller specimens dry more rapidly than dimensional lumber. In addition, none of the commonly used test methods simulate the accumulation of decaying organic debris that often occurs in connections of treated wood used above-ground [2]. Specimens are typically exposed in open areas to remove variability associated with natural shading, and when organic debris (leaf litter) does accumulate, it is removed during periodic inspections. In contrast, accumulation and decomposition of leaf litter is commonly observed in wooden structures, and it is possible that the presence of this decomposing organic matter increases the decay hazard to the adjacent wood product.

Tree cover has been identified as a major factor in understory biodeterioration and nutrient cycling in forest ecosystems [3,4,5,6,7] and leaf litter has been theorized to play a role in the rate of decomposition of coarse woody debris [8,9,10]. Although not previously reported, the same concepts could theoretically apply for residential wood structures located under the canopy of surrounding trees. It is possible that accumulation and persistence of leaf litter on the surface of residential above-ground structures can serve as both an inoculum source and a potential incubator for future fungal colonizers of the wood in contact [11]. Decomposing leaf litter may contribute to an increased decay hazard in at least two ways. Inadequate moisture is typically a limiting factor in fungal colonization of wood used above-ground [12,13,14] and is probable that the presence of the leaf litter slows drying of adjacent wood after rain events, thus increasing the proportion of time that the wood moisture content is conducive to decay. It is also possible that decomposing leaf litter plays a role, like soil, in providing a ready supply of nutrients and moisture that facilitates growth and sporulation of decay fungi [15,16,17]. The latter scenario is especially problematic because it suggests that preservatives evaluated for above-ground efficacy may need to provide protection in a broad spectrum of conditions, some of which are more similar to ground contact than currently assumed.

Although previous research has not directly evaluated the role of leaf litter in the decomposition of wood used above-ground, there have been reports of overlap between fungal groups associated with litter decomposition and wood decay [18]. Researchers evaluating the ability of leaf litter-degrading fungi to also degrade wood reported that many of the isolates caused some degree of weight loss in wood, although generally not to the extent caused by fungi traditionally associated with wood decay [19]. Hammel [20] notes that a succession of bacteria and fungi are thought to decompose leaf litter, with Basidiomycete fungi playing a role in lignin decomposition. Schneider et al. [21] also reported that, although more Ascomycetes were detected overall in leaf litter, Basidiomycetes did appear later in the degradation process, presumably because they were able to decompose remaining lignin compounds. It is thought that the Basidiomycete fungi that degrade leaf litter are more likely to be white than brown rot fungi, but this supposition has not been confirmed by research [20]. However, this white rot premise was circumstantially supported by a study which reported the Basidiomycete fungi found to be degrading leaf litter also caused the litter to have a bleached appearance [22]. Other researchers have reported a lack of Basidiomycetes, but those studies involved leaf litter in early stages of decomposition [23,24].

The available tools to characterize and observe fungal communities have increased dramatically in the last two decades. The development of next generation sequencing technologies has enabled large scale community level analysis and the resulting metagenomic capabilities allow researchers to analyze mixed microbial communities of interest and observe inter and intraspecific interactions [25,26,27]. Targeted microbial metagenomics, also referred to as metabarcoding or amplicon-based sequencing, is an extremely useful tool for dissecting complex and dynamic microbial communities that have been applied to forest soils [28,29,30], decaying wood [31,32], and standing trees [33].

The application of next generation sequencing (NGS) technology to characterize leaf litter is well represented in the literature, with less attention having been paid to processed lumber and the residential built environment. For example, Purahong et al. [34] reported dynamic shifts in fungal community composition as leaf litter ages/decomposes, where a general shift occurs from ascomycetous fungi to basidiomycetous fungi as the quality and composition of the leaf litter changes, which has also been reported by Zhang et al. [35]. Differences have been noted between the effects of litter in deciduous and evergreen forests [36], where plant diversity and litter biomass are key drivers in deciduous forest, but host effects outweigh these in evergreen forests. The impacts of global warming have also been studied using NGS paired with enzymatic assays [37] and the results of this and other studies found that decomposition rates did not accelerate with increasing temperature, but instead led to an increase in residual lignin paired with an increase of lignin degrading enzymes and increased presence of ectomycorrhizal fungi. The concept of home field advantage (HFA) is one that has garnered considerable interest in recent years [7,38,39], which states that plant biomass is more readily broken down in its native environment than a foreign one. This concept has not been tested in deterioration rates of processed lumber but would suggest that softwood species might degrade faster located in a predominant conifer overstory due to the prevalence of microbes adapted to breaking down the structural components of softwoods. This area represents a critical gap in our knowledge of wood decay in above-ground residential conditions and potentially challenges the current approach to wood protection.

An improved understanding of factors that affect the severity of above-ground decay hazards is critical to the development and evaluation of durable wood products. It is plausible that the presence of leaf litter may heighten the decay hazard for wood used above-ground by increasing wood moisture and/or serving as an inoculum source for wood decay fungi. The objective of this study was to increase our understanding of how decomposing organic matter contributes to decay in above-ground wood structures. In this study, we assessed the characteristics of young and aged litter types, their contribution to wood moisture content and decay, and utilized amplicon-based sequencing to identify and characterize the fungi found within young and aged leaf litter and adjacent to wood in an effort to compare the fungal communities of the leafy substrate to those which successfully colonize and ultimately degrade the wood.

## 2. Materials and Methods

### 2.1. Leaf Litter

The detritus that accumulates on above-ground structures could have a wide range of sources and characteristics depending on the type and proximity adjacent trees and shrubs [40], and one of the challenges of this study was selecting a characteristic or representative material. Two types of leaf litter were investigated, “aged” and “young”. The “aged” litter was a commercial product (Hsu organic STA Certified Leaf Compost) prepared in Wausau, Wisconsin, USA and available at garden centers in the Midwest. It has been composted for use as a soil amendment and has an appearance like coffee grounds. Product literature states that it is made from “tree leaves collected in the pristine woodlands of Wisconsin” (https://www.hsugrowingsupply.com/leaf-compost/hsu-leaf-compost). A compost analysis report was provided by the manufacturer (Table 1).

The other type of litter (young) was created from leaves (silver maple, sugar maple, elm, and white oak) that had been loosely piled outdoors for approximately 18 months in the Madison, Wisconsin area. The leaves were dried and then crushed to pass through a 6 mm (0.25 in.) screen. The intent of the young leaf litter was to evaluate the effect of litter in an earlier stage of decomposition than the commercial leaf litter. A compost analysis of the young leaf litter was conducted by the same laboratory that evaluated the commercial aged litter. The relatively high organic content, and the high carbon: nitrogen ratio and poor germination vigor are indicators that the young leaf litter had undergone less decomposition than the aged litter. Respiration was relatively low for the young leaf litter, but this is probably a function of the initial sterilization and subsequent dry storage prior to analysis. In contrast, the aged leaf litter had undergone compost analysis by the manufacturer prior to sterilization.

### 2.2. Treatment Groups Evaluated

Southern pine sapwood specimens were exposed under five conditions. One condition was without any preservative treatment and without leaf litter contact (Table 2). Comparison untreated specimens were exposed when placed in direct contact with either the aged or young leaf litter. In addition, preservative-treated specimens were exposed either with or without aged leaf litter contact. In this paper, discussion of the preservative-treated specimens is limited to the moisture content and decay evaluations.

### 2.3. Specimen Preparation and Exposure

All specimens were cut from southern pine (*Pinus taeda* L.) 38 by 89 mm (2 by 4 nominal) dimension lumber. The specimens were end-matched to minimize differences in moisture content and decay susceptibility associated with wood variability. One type of each specimen was cut from each of 5 “parent” boards (*n* = 25, specimens total, or 5 per treatment group). The lumber used for the untreated specimens was selected to be free of heartwood and other obvious defects. The preservative-treated specimens were cut from lumber that had been commercially pressure-treated with particulate copper azole at the target retention intended for above-ground use. Prior to exposure, all specimens were conditioned to uniform weight in a room maintained at 23 °C and 55% RH.

Two, 25 mm long stainless-steel screws were driven into each specimen 15 mm from one end to serve as electrodes for moisture content determination. The upper 13 mm depth of each hole was drilled to a larger diameter and filled with neoprene rubber sealant so that moisture measurements would be taken from the interior of the specimen.

A specimen holder was constructed to allow leaf litter to be trapped against a test specimen (Figure 1). The configuration approximately represents a moisture-trapping design in which the end of a deck board rests on doubled rim joists and butts against a fascia board. The specimen holders were constructed from 38 mm thick western redcedar lumber. Four drain holes were drilled through the bottom of the specimen holder. The specimens were placed flat in the bottom of the specimen holder, with 10 mm gap on all 4 sides of the test specimen. The designated type of leaf litter (if any) was then lightly packed into the gap around all 4 sides of the specimens until it was slightly below the upper surface of the specimen. Both the aged and young leaf litter were sterilized by autoclave prior to use to eliminate existing fungal growth. The specimens/holders were placed onto an above-ground rack at a test site west of Madison, Wisconsin, USA in June of 2012. Shade cloth (50% shading) was stretched over the rack to simulate the shading that might occur in areas of leaf litter deposition.

### 2.4. Specimen Evaluations

Moisture Content: Specimen moisture content was evaluated on an approximate weekly basis using a General Electric Protimeter Timbermaster (Amphenol, St. Mary, PA, USA), resistance type moisture meter. The internal calibration recommended for southern pine was used in this study. Readings were taken by contacting the meter pins with the stainless-steel screws that had been inserted into each specimen. Although the accuracy of resistance type moisture meters declines above the fiber saturation point, recent research has shown that resistance moisture meters can provide useful information on moisture contents above the fiber saturation point when screws are used as the electrodes [2]. Readings were adjusted for wood temperature as described in Lebow and Lebow [2]. Moisture measurements were not conducted during freezing temperatures as initial attempts indicated that readings taken on frozen wood underestimated moisture contents of specimens above the fiber saturation point.

Visual Decay Evaluations: After 4, 13, 24, and 41 months of exposure, the specimens were removed from the holders, brushed free of leaf litter (if applicable), and visually examined for evidence of fungal decay. They were assigned a condition rating patterned after that described in the American Wood Protection Association (AWPA) Standard E18 [1,41] (O = failed, 10 = sound, with ordinal ratings 9–4 based on percent removal of wood cross section due to decay). The specimens were then returned to the holders and re-packed with the original leaf litter plus any additional litter needed to bring up to the original depth (if applicable). A visual example of an untreated pine block at the end of the test is presented in Figure 2.

Comparisons of the visual ratings for the different treatment groups were based on a cumulative logit model estimated with SAS^®^ V9.4 (SAS Institute Inc., Cary, NC, USA) procedure GLIMMIX with main effects for treatment groups and exposure time and a random effect for specimens to capture dependencies for repeat measurements over time.

### 2.5. Amplicon-Based Sequence Analysis

Amplicon-based sequencing analysis of the microbial community associated with both specimens and leaf litter was also conducted at two time points. After 25 and 41 months of exposure, selected samples of leaf litter and of wood from the specimens were collected for amplicon-based sequencing analysis. Amplicon-based sequencing was based on 4 replicates of each treatment group. Wood samples were obtained by drilling into the bottom of specimens 13 mm from the end grain and 13 mm from an edge of the wood. For simplification and cost effectiveness, only leaf litter and wood samples from untreated wood were analyzed. Samples of unexposed young and aged litter controls were included at both time points. Leaf litter samples were frozen at −30 °C for approximately 1 month before processing. Samples were mixed by hand in the plastic sample bag, 0.25 g was weighed out and DNA extracted using the MoBio Power Soil DNA Isolation Kit (Qiagen, Germantown, MD, USA). The 100 µL DNA solutions were then cleaned using the MoBio Powerclean Pro DNA Clean-up Kit (Qiagen, Germantown, MD, USA), then quantified by Biotek spectrophotometer (Biotek, Winooski, VT, USA) and diluted to 10 ng/µL in 10 mM Tris 1 mM EDTA (TE, pH 8).

Sawdust was frozen at −30 °C for approximately 1 month before processing. Samples were mixed by hand and 0.1 g was added to 800 µL 2% CTAB buffer with 0.1% beta-mercapto-ethanol and ground for 30 s with a hand drill and plastic pestle. Samples were then incubated 1 h at 65 °C and centrifuged 15,000× *g* for 3 min. Supernatants were transferred to spin columns from the Promega Wizard SV Genomic DNA Purification Kit (Promega, Madison, WI, USA) and manufacturer instructions for purification were followed. Samples were re-suspended in 100 µL water with RNAse inhibitor as recommended then diluted to 2 ng/µL in TE pH 8.

Twenty-five nanograms of leaf litter DNA and 5ng wood DNA samples were amplified in triplicate by PCR using ITS1F (CTTGGTCATTTAGAGGAAGTAA) and ITS2 (GCTGCGTTCTTCATCGATGC) primers with Illumina adapters for the MiSeq platform (Illumina, San Diego, CA, USA) and 22 unique identifiers on the reverse primers. The amplified region of interest is the internally transcribed spacer region 2 (ITS2) as described in De Gannes et al. [42]. Phusion Hot Start Flex DNA Polymerase (New England Biolabs, Ipswich, MA, USA) in HF buffer was used for PCR’s with the following program: 4 min at 94 °C, followed by 30 cycles of 30 s at 94 °C, 60 s at 50 °C, and 90 s at 72 °C and a final extension of 10 min at 72 °C. A 400 bp product was confirmed on 2% Agarose gel electrophoresis. Each set of replicates was combined and cleaned up with Agencort AMPure XP beads (Beckman Coulter, Indianapolis, IN, USA) following manufacturer instructions. Cleaned samples were quantified using the Quant it DNA Assay Kit (high sensitivity, Invitrogen, ThermoFisher Scientific, Waltham, MA, USA) using the microplate procedure and the Biotek Synergy H1 multimodal plate reader (Winooski, VT, USA). Samples were normalized to 10 nM and combined, then submitted to the University of Wisconsin-Madison Biotechnology Center–DNA Sequencing Facility for Illumina MiSeq sequencing.

Amplicon-based sequencing data were processed using the AMPtk v1.3 pipeline. AMPtk is a series of scripts to process NGS amplicon data using USEARCH and VSEARCH, it can also be used to process any NGS amplicon data and includes databases setup for analysis of fungal ITS, fungal LSU, bacterial 16S, and insect COI amplicons. It is compatible with Ion Torrent, MiSeq, and 454 data [43]. For this analysis, overlapping 2 × 250 bp Illumina MiSeq reads were merged using USEARCH9 [44], forward and reverse primers were removed from the merged reads, and the reads were trimmed or padded with N’s to a set length of 250 bp. Operational taxonomic units (OTUs) were generated for each sample, which is used to describe taxonomically distinct groups of fungi [45]. Processed reads were quality trimmed based on accumulation of expected errors less than 1.0 [46] and clustered using the UPARSE algorithm using default parameters (singletons removed, 97% OTU radius). An OTU table was generated by mapping the original reads to the OTUs using VSEARCH 1.9.1 [47] and the OTU table was subsequently filtered to eliminate “index-bleed” at 0.5%. Taxonomy was assigned using a combination of UTAX and global alignment (USEARCH [44] to the UNITE database [48]) and non-fungal OTUs were removed prior to downstream data processing.

### 2.6. Community Analysis and Species Richness Analysis

PCORD 7.29 (MjM Software Design, Gleneden Beach, OR, USA) [49] was used to perform community analysis to provide more quantitative information on specific relationships within the data set. Lack of fit was evaluated based on PCORD’s stress and instability measurements. Output OTU tables from the previous section were imported and used to address the following questions:

1. Does time of exposure to leaf litter impact fungal colonists in the wood (25 vs. 41 months)? Comparisons performed on wood only—removed all leaf litter from the dataset. Fungal matrix had OTUs occurring less than 10 times removed (total of 677 OTUs analyzed). Fungal matrix was relativized by sample unit to standardize sampling depth. Groupings were made of wood from each sample period (25 months, 41 months) exposed to each litter type (no litter, aged, and young) resulting in six factorial treatment groups of interest. Nonmetric multidimensional scaling (NMDS) ordinations and multi-response permutation procedures (MRPP) were performed using the Sorensen distance measure for both. Group comparisons of interest for this question included 1v4, 2v5, 3v6. Additionally, groups 1–3 (25 months) and 4–6 (41 months) (Table 2) were combined for an additional MRPP analysis to look at the effect of year on wood fungal communities.

2. How well does community structure match from litter to wood? Comparisons performed on all samples, both leaf litter and wood. Fungal matrix had OTUs occurring less than 10 times removed (total of 2223 OTUs analyzed). Fungal matrix was relativized by sample unit to standardize sampling depth. Groupings were made of each leaf litter type (aged, young) and wood exposure type (aged litter, young litter) were compared between sampling periods (25 months, 41 months) for a total of eight exposure scenarios to compare similarity between the leaf-litter to the wood to which it was exposed. Non-metric multidimensional scaling (NMDS) and MRPP were performed using Sorensen distance measure for both. Group comparisons of interest for this question included 1v5, 2v6, 3v7, 4v8.

3. Were there any differences between aged and non-aged litter? These analyses were run on the whole dataset.

In addition to the aforementioned analyses, indicator species analyses (ISAs) [50] were used to detect, and describe the significance of, fungal taxa indicative of a priori treatments [51]. Due to multiple group-comparisons, only highly significant taxa were included. Additional tests of species richness were also performed in PC-ORD to determine the contributions of organic detritus to species richness when placed in contact with wood. Diversity measures were calculated using the following formulae:
*S = Richness = total count of non-zero elements in a row,*(1)
*E = Evenness = H/ln (S),*(2)
*H = Diversity = sum (Pi*ln (Pi)),*(3)
*D = Simpson’s diversity index = 1− sum (Pi*Pi), where Pi = importance probability in element I (element I relativized by row total).*(4)

### 2.7. Functional Guilds Analysis

To provide additional contextual information on the fungal species from this study, OTUs from ampTk were further processed using Funguild [52], an online tool for characterizing fungal species within a community based on their ecological roles. Funguild builds on taxonomy data in the output OTU table to include ecological function for each identified fungal species and classifies them based on biological function (saprophyte, parasite, endophyte, etc.), wood saprobes are also characterized in Funguild and decay type is typically indicated as brown, white, or soft rot fungi. The goal of the guilds characterization was to determine if different groups of fungi predominate when leaf litter sources were varied or absent and to determine if the presence of absence of leaf litter had any impacts on the guild composition which could in turn affect rates of wood decay.

## 3. Results

### 3.1. Specimen Moisture Content

As expected, contact with leaf litter was associated with higher moisture contents in all the untreated and wood specimen and this effect was observed with both the aged and young leaf litter for the untreated specimens (Figure 3). Within 6 months of exposure, untreated specimens in contact with leaf litter had moisture contents consistently above 30%, and moisture contents were above 40% for the vast majority of the exposure period. Even the specimens exposed without leaf litter often had moisture contents above 30% by the second year of exposure. Average moisture contents were slightly lower in the treated specimens, but those in contact with aged leaf litter typically had moisture contents above 30% and often had moisture contents above 40%.

### 3.2. Visual Decay Evaluations

The specimens placed in contact with aged leaf litter exhibited more evidence of fungal decay than did specimens not placed in contact with leaf litter (*p* = 0.0002), as did specimens placed in contact with young leaf litter (*p* = 0.0005; Figure 4). After 41 months of exposure, one of the untreated specimens exposed to aged leaf litter was so decayed that it crumbled upon removal from the specimen holder and was rated as a “0” on the adopted AWPA rating scale. This specimen also had a carpenter ant infestation. A second untreated specimen exposed to aged leaf litter was sufficiently decayed to rate a “6” on the AWPA rating scale. Although the aged leaf litter specimen ratings were not statistically different than those for the young leaf litter across time (*p* = 0.3100), they were statistically different at 41 months (*p* = 0.0003). One of the treated specimens exposed to the aged litter (not coincidentally a specimen with low preservative retention) also had clear evidence of decay along one edge. Decay was less obvious in specimens in contact with the young leaf litter, but two untreated specimens did have some decay. Only one untreated specimen not in contact with leaf litter showed slight evidence of decay, although a second specimen was considered to have “possible” early stages of decay.

### 3.3. Amplicon-Based Sequencing Analysis

A total of 3352 fungal OTUs were recovered from the metagenomic analysis. Taxonomically, 1948 of the recovered OTUs were Ascomycetes, 537 were Basidiomycetes, 162 Glomeromycetes, with fewer representatives of the Mucoromycetes or other relevant species. In addition, 361 Agaricomycete OTUs were recovered, these include the mushroom forming fungi that comprise most of the commonly known decay fungi. Fifty OTUs classified as belonging to the order Polyporales were recovered. A complete OTU table with annotated taxonomic designations for all samplings is available as Appendix A on the MDPI website. The OTU table generated using AMPtk was imported for further community analysis using PC-ORD and Funguild [40]. Amplicon-based sequencing data has been archived at the National Center for Bitoechnology Information (NCBI) sequence read archive (SRA) under Bioproject# PRJNA612060.

### 3.4. Community Analysis and Species Richness Analysis

The results of the community analysis showed differences between treatments based on the fungal species composition in wood both with and without leaf litter and across sampling intervals. Combined for both years, aged leaf litter had a distinctly different fungal community than young leaf litter (MRPP *p* = 0.0001 A = 0.123). Significant differences were noted between aged and young litter and wood exposed to either young or aged litter. The results from the community analysis are summarized in Table 3 and contain references to proceeding Non-metric Multi-dimensional Scaling (NMDS) ordinations as well as contain lists of indicator species associated with comparisons of interest.

To observe the effects of leaf litter on fungal colonists over time (25 vs. 41 months), multi-response permutation procedures (MRPP) were performed on wood samples only and compared wood from 25 months (25 mos.) exposed to aged, young, and no leaf litter to wood from 41 months (41 months) subject to similar exposure. MRPP was deemed to be the more suitable option for analysis in this case due to uneven sample numbers, which are not ideal for a more robust analysis, such as permutational multivariate analysis of variance (PERMANOVA). The results (Figure 5A) indicate that wood with no litter (Wood14N and Wood16N) were more similar in species composition than those exposed to either aged (A) or young (Y) litter. NMDS axis 1 accounted for 34.7% of the variation, NMDS axis 2 accounts for 19.6% of the variation, and axis 3 (not shown) accounted for 13.4% of the variation (Stress = 11.998, Instability = 0.0000, and *p* = 0.004).

In order to test overall similarity in community structure from litter to wood, comparisons were made of all samples, both litter and wood and grouped as such: (A) aged leaves 25 months, (B) young leaves 24 months, (C) aged leaves from 41 months, (D) young leaves from 41 months, (E) wood from 24 months with aged litter, (F) wood from 24 months with young litter, (G) wood from 41 months with aged litter, and (H) wood from 41 months with young litter. NMDS and MRPP were performed using Sorenson distance measure and grouped comparisons were made comparing A-E, B-F, C-G, D-H to evaluate the similarity between: (A–E) aged leaves versus wood in 25 months (Not different, *p* = 0.094, A = 0.06), (B–F) young leaves versus wood in 25 months (Different, *p* = 0.004, A = 0.102), (C–G) aged leaves versus wood in 41 months (Not different, *p* = 0.187, A = 0.033), and (D–H) young leaves versus wood in 41 months (Different, *p* = 0.011, A = 0.115). The resulting NMDS ordination is presented in Figure 5B. Young and aged leaf litter controls (14CY, 16CY and 14CA, 16CA) at the two time points remain nearly identical in community structure, indicating that there was similar sequencing coverage between time points. The communities’ group by leaf type and leaf samples (right side of figure) separate from wood samples (left side of figure). Young leaf samples were significantly different from their paired wood at both time points, which could be an indication of early leaf colonizers that may not be able to establish in wood. Other pairings were not significantly different, indicating there may be some carry over directly from aged litter to wood.

Comparing only wood fungal communities without regard to litter treatment groups showed little community differences between wood samples in 25 months and 41 months (MRPP; *p* = 0.039; A = 0.017). Based on these results, it is suggested that fungal species composition was mostly similar after 24 and 41 months of above-ground exposure when looking at the wood only (Figure 5C). It should be noted that there were far fewer OTUs detected in wood vs. leaf litter, so this likely has a large influence on this data set. Although the *p*-value was significant, the low A-value indicative of within-group heterogeneity being expected by chance led to our conservative assessment of wood fungal communities.

With wood removed from the dataset (looking at litter only), there are still clear differences between young and aged leaf litter (MRPP *p* = 0.009, A = 0.126) regarding fungal species composition (Figure 5D). However, over time, litter species composition stayed relatively similar between 25 months and 41 months for both young (MRPP *p* = 0.677; A = −0.026) and aged (MRPP *p* = 0.651; −0.031) leaf litter.

The results of the indicator species analysis (ISA) are presented previously in Table 3. Indicator species are compiled for several comparisons of interest made during the earlier community analysis. Indicator values were only deemed relevant below the *p* = 0.01 level and a calculated indicator score of 60% or higher (*p* < 0.01; IV ≥ 60%). As presented in the table, each question required a unique ISA to elucidate ecologically relevant indicator taxa for the comparisons of interest. In each category, the maximum indicator value is an indicator of how often each species occurs in the highlighted condition. 

As a final metric of the fungal community, species richness was calculated for each exposure scenario and the results of the richness analysis are shown in Table 4. No significant differences were noted between evenness, or 2 independent measures of diversity. Some differences were noted in species richness with highest species richness observed in aged leaf litter and lowest species richness observed in wood in contact with aged litter, but this was not significantly different from wood not exposed to leaf litter. Moving the analysis from leaf litter into solid wood presents a significant bottleneck as fungal colonization is limited by space and available moisture. 

### 3.5. Guild Descriptions

Out of a total of 3213 OTUs, 2070 were assigned guild information using the Funguild [52] software. A complete OTU table with annotated guilds data is presented as Table 2. A total of 257 OTUs were classified as animal pathogens, these are not discussed here as they are likely not involved in the breakdown of woody biomass. A total of 144 OTUs were classified as dung saprotrophs (DSAP), mostly characterized as having broad host ranges and occurring on soil, grass, dung, or rotten wood. Sixty-four OTUs were classified as ectomycorrhizal fungi (ECTOMY) of which one was also classified as a white rot (*Sistotrema brinkmannii*). Twelve OTUs were classified as bryophyte parasites (BRYPAR) (various species of *Pluteus*, *Galerina*, and Hymenoscyphus). Only 8 OTUs were classified as arbuscular mycorrhizae and were only identified to the order level (Diversisporales, Gigasporales, Archeosporales, and Glomerales), all within class Glomeromycetes. Pie charts indicating the guild composition of each exposure scenario is presented in Figure 6. A general shift was noted as the litter ages, with the guild composition moving from predominately soil saproptrophs to a more balanced composition that included higher percentages of litter saprotrophs, wood saprotrophs, and fungal pathogens.

A total of 353 OTUs were classified as wood saprotrophs (WSAP). These were more prevalent in the litter compared to the wood and between the wood samplings, more were detected in the 41-month sampling. Among the wood saprotrophs, 86 OTUs were described as having traits indicative of soft rot, 63 were described as having traits indicative of brown rot, 54 were described as having traits indicative of white rot, 3 were classified as being either brown or white, but this was due to only being described to the family taxonomic level. The remaining 151 were classified as NULL meaning a functional trait could not be determined. The NULL group was a mixture of microfungi with agaricoid, gasteroid, tremelloid, or phalloid growth morphologies and represent fungi in the database that have not yet been assigned trait information.

Brown rots were detected in both leaf litter and wood and the predominant species identified was *Dacrymyces capitatus*. *Rhodonia placenta* was also detected in wood, but only in the 24-month leaf and wood samples. White rots were much more diverse in both the leaf litter and the adjacent wood, and the most common white rot fungal genera identified were *Irpex*, *Phlebia*, *Phaeanerochaete*, *Gleoporus*, *Ceriporia*, *Sistronema*, *Trametes*, and *Peniophora*.

## 4. Discussion

### 4.1. Specimen Moisture Content

It is likely that the lower moisture contents in the treated specimens reflect differences in wood properties rather than an effect of the preservative treatments. Specimens cut from one of the treated parent boards contained heartwood, and these specimens consistently produced relatively low moisture content readings. In general, the moisture readings indicate that the specimens placed in contact with leaf litter had moisture contents conducive for the growth of decay fungi [53,54,55,56]. Decay fungi require a moisture content of at least 20% to sustain any growth, and higher moisture contents (over 29%) are required for initial spore germination [14,57]. For optimal growth, brown and white rot decay fungi typically prefer wood in the moisture content range of 40–80% [58,59]. Soft rot fungi, however, tend to prefer these conditions for colonization and growth [60], which can also severely decay wood under these conditions.

### 4.2. Visual Decay Evaluations

The aged leaf litter contributed to decay development in the untreated specimens and did so to a greater extent than the young leaf litter. There are at least four possible mechanisms for the aged litter promoting decay. One is moisture entrapment and the elevation of specimen moisture content to a range more conducive to fungal growth [14]. Another is that the litter served as the germination site for inoculum that subsequently colonized the wood [61]. A third possibility is that the fungi initially became established in the wood but benefited from nutrient, extractives, or lignified residues that diffused into the wood from the leaf litter [62,63,64]. A fourth possibility is that there are simply fungi present that can decompose both litter and wood [34]. Moisture content does not appear to be the sole mechanism because the specimens in contact with young leaf litter had similar moisture contents but exhibited less evidence of decay. Some of the specimens not in contact with any leaf litter also maintained relatively high moisture contents.

### 4.3. Amplicon-Based Sequencing Analysis

The results of this study agree with the basic findings of many studies that have focused on the dynamics of fungi in leaf litter [8,11,34,35,62,65,66,67,68]. As leaves accumulate, they typically contain a lot of leaf and soil saprophytes, that can readily break down the leafy debris into less complex and more nutritionally devoid substances [69,70,71,72,73,74]. As the composition of the leaf litter changes to more closely resemble soil, the composition of fungi found in the litter shift and reflect a more specialized consortium of fungi that can readily exploit more nutrient poor resources [35], which is the situation in wood. These fungi are those that are able to exploit the basic structural components (cellulose, hemicelluloses, and lignin) that make up the recalcitrant portions of the leaf litter (petioles, branches, twigs, etc.) as well as any non-durable wood that comes in contact with the leaf litter. Lignin content has been the focus of several studies as a rate limiting factor in decomposition and nutrient cycling, where proportional shifts in more the recalcitrant compounds serves to throttle litter decomposition as a conservation strategy to prevent depletion of soil nutrients [32,62,64,75,76] and would also theoretically select for fungi that are able to break down lignin, i.e., white rot fungi [11,63]. The compositional shift that was noted in our study has been reported by several additional studies [35,77,78] and others, where a gradual replacement of ascomycetous fungi (leaf saprobes) with more basidiomycetous (wood saprobe fungi) due to the changing litter composition, also noted in Zhang et al. [35]. This result highlights the importance of routine preventative maintenance of wood in above-ground exposures because over time the fungal composition will shift, and the result will be higher inoculum loads of wood decay basidiomycetes present in close contact with the wood surfaces.

### 4.4. Community and Richness Analysis

When comparing litter between samplings (Age Combined, Table 3), a total of six fungal species were found to be significantly associated with the leaf litter after 41 months of test exposure (*Sistotremastrum guttuliferum*, *Peniophorella pubera*, *Peniopherella praetermissa*, *Arrhenia* spp., *Rhodonia* (*Postia*) placenta, and another species of *Sistotremastrum* only identified to the genus level). *Peniophorella praetermissa* was identified as a prominent white rot isolated by Clausen and Lindner when looking at performance of shaded pine and maple lap joints, also in Madison, WI [79]. When comparing the aged to young litter (Years Combined), three fungal species were found to be associated with the aged litter (*Ganodermataceae* sp., *Coprinellus* sp., and *Dacrymyces capitatus*). Several species of *Dacrymyces* are occasional rots on wood including external wood work [80], window framing [81], or spruce shingles [82], but all seem to be overly wet environments similar to this study. Fruiting bodies of fungi resembling *Dacrymyces* sp. were photographed on samples at the end of this study (Figure 2), so these results were consistent with conditions observed during the test. When data are separated by year (Years Separate), *Polyporus* spp., *Hebeloma* spp., *Hyphodermella* spp., *Corinopsis* spp., and *Lepista* spp. were all found to be significantly associated with aged litter after 25 months of field exposure, while *Peniophera* spp. was found to significantly associated with the aged litter samples after 41 months exposure. When comparing the fungal taxa only in the wood (Age Combined), *Sistotrema* spp. was found to be significantly associated with wood samples after 25 months exposure while *Subulicystidium brachysporum*, *Arrhenia* sp., and *Rhodonia placenta* were found to be significantly associated with wood samples after 41 months exposure. *R. placenta* is commonly used in laboratory assays to evaluate decay resistance and has been shown to modify lignin in decayed samples [76], the remainder of these fungi are commonly found on late state coarse wood debris.

It is worth noting the abundance of ectomycorrhizal (ECM) fungi found within each of the above comparisons which indicates ECM fungi may contribute to the overall litter and wood decay fungi processes although they are not always considered as such. ECM fungi were also abundant in previous samplings by Kirker et al. [83] looking at soil fungal communities in soils subjected to long term preservative exposure. Prior studies have also noted the importance of ECM fungi in the overall decay process [78,84,85,86] as well as potential soil bioremediation tools [87,88,89,90].

The overall results of these analyses indicate that aged organic detritus appears more similar to and likely contributes a fair amount to the species composition of the adjacent wood and that young and aged litter develop distinct fungal communities. Young litter likely promotes growth and establishment of more litter decomposing fungi than wood decay fungi.

### 4.5. Guild Descriptions

As noted previously, a general shift in guild composition was noted as the litter aged and was also observed in wood exposed to aged litter. It is suspected that this shift is in response to the changing nutritional value of the substrate which would in turn select for a higher proportion of fungi that can breakdown the remaining woody biomass, which would also agree with the findings of Purahong [34,35,91,92] and Zhang et al. [35] demonstrating shifts in community composition as the substrate is altered. The majority of the OTUs classified as dung saprotrophs were isolated from the leaf litter, but some were detected in wood as well (*Preussia pilosella*, *Sprorormia subticinensis*, *Sordaria fimicola*, *Chaetomium globosum*). As noted previously in the results, one of the detected fungal OTUs is considered to have both ectomycorrhizal characteristics but is also often characterized as a white rot fungus. *S. brinkmannii* is considered a weak white rot fungus and is often followed in succession with more late stage wood rot fungi such as *Stereum hirsutum* or *Bjerkandara adusta* [93]. Interestingly, *S. brinkmannii* may be capable of both significant wood decay and for ectomycorrhizal associations with both conifers and hardwoods [94,95]. Since there was only one specimen that rotted to the point of failure on our rating scale after 41 months, most of these samples would be considered at early stages of decay. Relatively high diversity of ectomycorrhizal fungi were detected in the leaf litter (both young and aged) and representatives of corticoid (ex. *Sistoterma*, *Tulasnella*, and *Tomentella*), agarocoid (ex. *Tricholoma*, *Hebeloma*, *Russula*, *Cortinarius*), boletoid (ex. *Suillus* spp. and *Fuscoboletinus* spp.), *clavaroid* (ex. Thelephora spp.), and *gasteroid* (ex. Scleroderma, Tuber, and *Elaphomyces* spp.) basidiomycete fungi were detected, as well as numerous microfungi, which were only classified to the family level (*Helotiaceae* and *Ceratobasidaceae*). Ectomycorrhizal fungi are typically considered symbionts in the soil environment and not considered active wood decay fungi, however previous studies have shown that they are an integral part of nutrient cycling in forest soils and leaf litter [96,97,98].

Comparing brown rots to white rots, much less diversity was noted in the brown rot guilds compared to the white rot guilds. This is likely due to the composition of the substrate; the leaf litter was composed of mostly hardwood leaves and would theoretically support growth of white rot fungi over brown rots and the data bear this out to some degree [99,100,101,102]. There were noticeable temporal differences in certain fungi, but these should not be attributed to early or late successional histories based on our limited sampling; further studies would be needed to substantiate these in a meaningful way. Soft rot fungi were more prevalent in the leaf litter and sporadic in the wood but did increase in numbers between the 24- and 41-month wood samples. Among the unassigned or NULL trait group, the majority of these OTUs were only found in the leaf litter samples and only a few OTUs (*Fusarium*, *Cladopsorium* and *Orbilia* spp.) were detected in the wood samples. A notable exception was *Simocybe sumptuosa*, which is a brown spored agaric in the Crepidotaceae family that has been previously isolated from *Picea* logs in Norway [103]. This species was detected in the 24-month wood samples but not the 41-month wood samples. The diverse pool of wood saprophytes detected in this study give clear indication that the buildup of organic detritus on and adjacent to wood above-ground provides a ready source of fungal inoculum and that many of these fungal species can readily colonize wood given the proper conditions. The wood samples processed in the genomic portion of this study were untreated southern pine and additional studies are on-going to understand this process in wood that is chemically treated.

## 5. Conclusions

The purpose of this study was to demonstrate that leaf litter presents a ready source of fungal inoculum for wood above-ground and can negatively impact the performance of untreated wood. The results of this study highlight several important factors for consideration: The first is that the buildup of organic detritus contributes to the moisture accumulation and subsequent moisture content of adjacent wood. The second is that aged litter in contact with wood in above-ground scenarios contributes to increased wood decay over the 41-month exposure. Additionally, organic detritus contributes to the above-ground decay process by providing a large and diverse assemblage of fungal inoculum including wood decay fungi. Over time, the proportion of wood decay fungi increases as the litter ages and compositionally becomes more similar to soil. If this is applied to an above-ground decking situation, years of accumulation of leaf litter can collect in inaccessible areas and the result is potentially ground contact decay pressure in a very localized area. This can affect not only the surface decking, but also the adjacent sub-structural elements, such as deck joists and ledger boards. Although the fungal community established in the young litter was significantly different from the adjacent wood, there was similarity between the aged litter fungal community and the wood exposed to the litter at both time points. Communities detected in both leaf litter types in 24 months remained in the litter at 41 months indicating that the organisms persist once established. Similarly, those fungi detected in wood at 24 months were similar to those detected at 41 months. The unexposed wood (no litter) was the least similar suggesting that without litter there was less inoculum available in proximity to the wood and inoculum may come from outside sources. A shift in community and guild composition between aged and young litter that persisted through subsequent samplings. While ascomycetes predominated the earlier exposures, more basidiomycetes were present in the aged litter. The results of this study raise an important consideration when protecting wood exposed above-ground: If leaf litter is not routinely removed from the wood surface, a more severe decay risk may be present than prescribed in current building code designations. Areas where leaf litter is likely to collect and not accessible to routine maintenance may require wood preservative retentions intended for ground contact scenarios. Future studies will build on this concept and use a similar approach to understand these exposure scenarios at different weathering sites and when leaf litter is exposed to chemically treated wood.

## Figures and Tables

**Figure 1 microorganisms-08-00696-f001:**
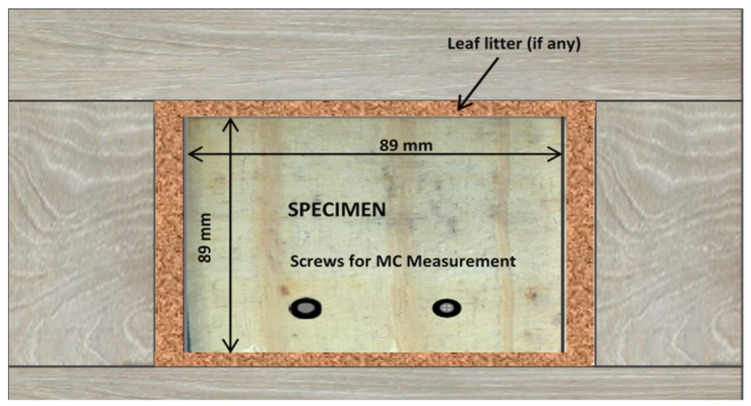
Physical configuration of specimen holder used in this study. The wood block was surrounded by either aged or leaf litter and allowed to weather. Black holes near the bottom were used to obtain moisture content at each observation. The assembly shown represents one replicate.

**Figure 2 microorganisms-08-00696-f002:**
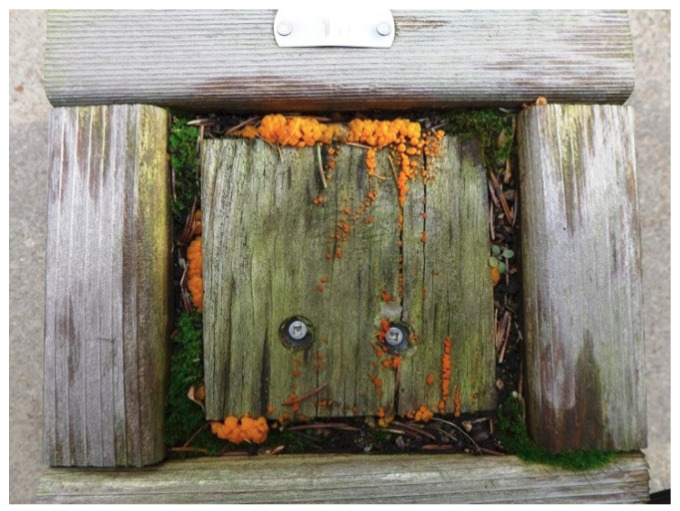
Visual example of untreated pine block at the end of the test. Note moss growth indicative of high moisture content and surface growth of saprophytic fungi on the surface of the block, presumably *Dacrymyces* spp. (likely *capitatus*).

**Figure 3 microorganisms-08-00696-f003:**
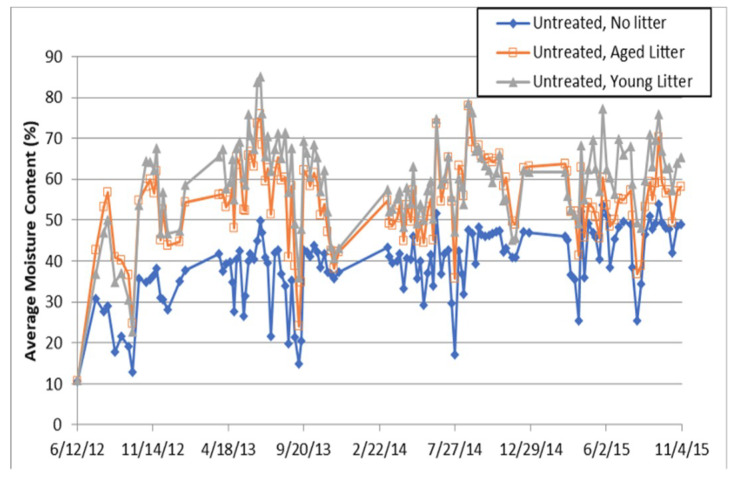
Average moisture content for unexposed wood blocks compared to wood exposed to aged and young leaf litter (*n* = 5). Note higher swings in moisture content when leaf litter is present and slightly higher moisture content (MC) in aged vs. young litter. Error bars omitted for readability. Average standard errors were 2.4%, 4.20%, and 3.63% for no leaf litter, aged leaf litter, and young leaf litter, respectively.

**Figure 4 microorganisms-08-00696-f004:**
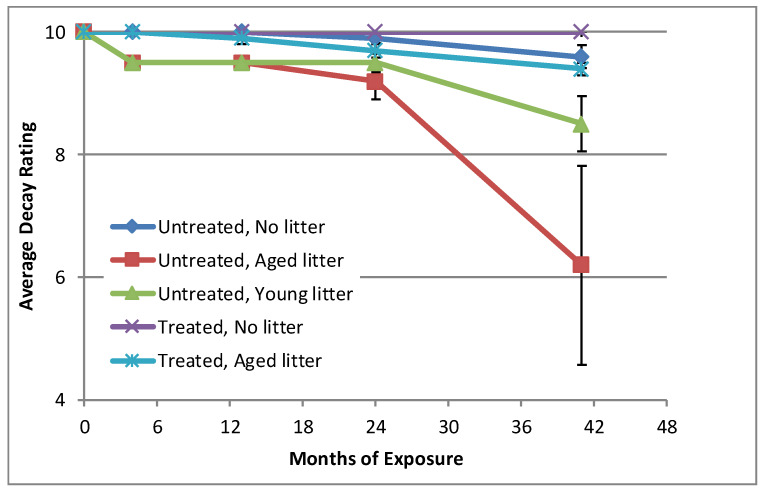
Average decay rating (*n* = 5) of above-ground wood samples in different exposure scenarios. For comparative purposes, treated wood both with and without litter are included in this figure but not discussed in the manuscript. Note faster rate of decline in decay rating of untreated wood in contact with aged litter compared to both young litter and litter free exposure scenarios.

**Figure 5 microorganisms-08-00696-f005:**
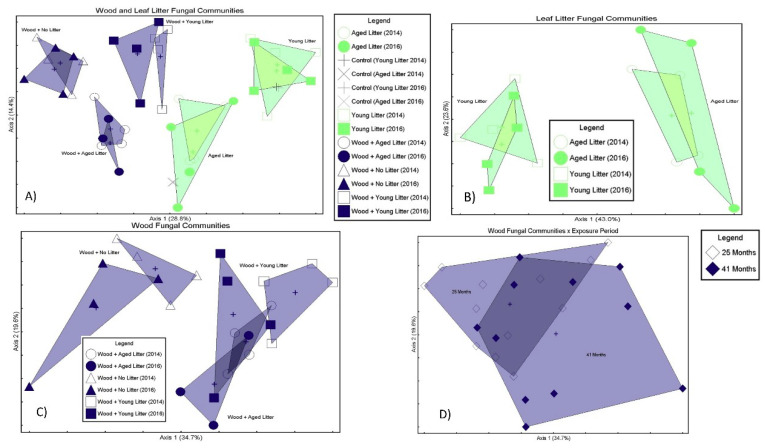
NMDS ordinations depicting: (**A**) Similarity between wood fungal communities exposed to aged (A), young (Y), or no leaf litter (N); (**B**) similarities between all treatment groups: unexposed controls (lf14-lf16HM (young) and lf-16SB (aged) compared to exposed leaves leaf14–16Y (young) and leaf14–16A (aged) and finally wood exposed to either young (Y) or aged (A), as well as no litter (N). (**C**) All samples grouped, overall similarity between fungal communities at the two exposure periods 2014 (25 months) and 2016 (41 months), showing similarities in the fungal communities. (**D**) NMDS of fungal communities in litter only showing differences in young (Y) and aged (A) litter that persisted over the two sampling times.

**Figure 6 microorganisms-08-00696-f006:**
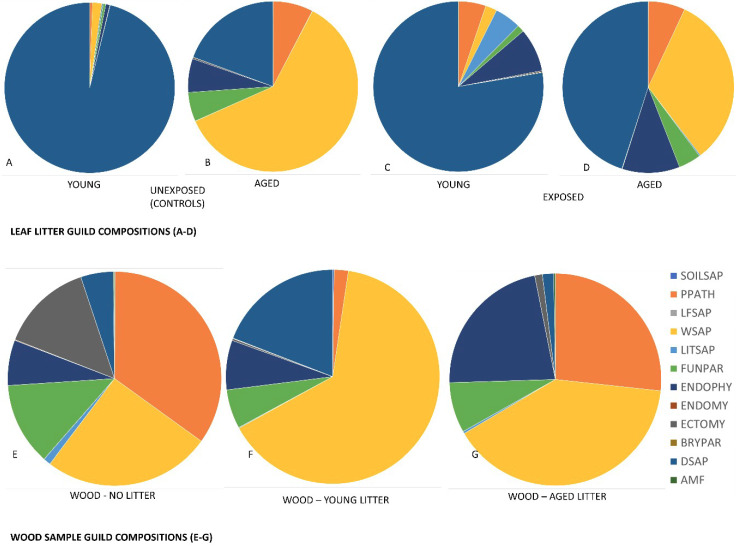
Guild composition of fungi under 7 different exposure scenarios. (**A**) Young litter prior to exposure, (**B**) aged leaf litter prior to exposure, (**C**) young litter after exposure, (**D**) aged litter after exposure, (**E**) wood with no leaf litter in contact, (**F**) wood exposed to young leaf litter, and (**G**) wood exposed to aged leaf litter. Note differences in relative proportion of fungal guilds and increased proportion of wood saprophytes (WSAP) between young and aged litter and wood exposed to both young and aged litter. Functional guild assignments performed using FungGuild [52]. Figure legends to the right are defined in the text.

**Table 1 microorganisms-08-00696-t001:** Characteristics of the two types of leaf litter evaluated. Nutrient analysis conducted by A & L Great Lakes Laboratories, Fort Wayne, IN, USA. ^a^ Expressed as mg carbon (from CO_2_) per gram per day. TS = total solids, OM = organic matter. ^b^ Respiration may have been reduced by earlier autoclaving and subsequent dry storage.

Litter Characteristic	Aged	Young
Nitrogen (%)	1.78	1.63
Phosphorous (%)	0.09	0.17
Potassium (%)	0.26	0.51
Magnesium (%)	0.5	0.51
Calcium (%)	1.78	3.5
pH	7.9	6.5
Organic matter (%)	48.58	81.84
Organic carbon (%)	24.29	40.92
Carbon: Nitrogen (C/N) ratio	13.6:1	25.1:1
Germination emergence (%)	90	100
Germination vigor (%)	71	15
Respiration, TS ^a^	1	5 ^b^
Respiration, OM ^a^	1	1 ^b^

**Table 2 microorganisms-08-00696-t002:** Treatment groupings used for field studies and subsequent analyses in this study.

Group	Treatment	Time (mos)
1	No Litter	25
2	Aged Litter	25
3	Young Litter	25
4	No Litter	41
5	Aged Litter	41
6	Young Litter	41

**Table 3 microorganisms-08-00696-t003:** Community analysis results grouped with Indicator species analysis results. Grouping and treatment indicate exposure scenarios of interest while Figure relevance references resulting Non-Metric Multidimensional Scaling (NMDS) ordinations produced from the data. Lastly, indicator species derived from those exposure scenarios are listed by treatment group. ^a^ IV = Maximum indicator value, ^b^
*p*-value = significance level; *p* < 0.01 indicates highly significant indicator taxa for respective treatments.

Grouping	Treatment	Figure Relevance	Taxon	Observed Indicator Value (IV) ^a^	*p*-Value ^b^	OTU #
Litter 25 vs. 41 (Age Combined)	Litter 41	Figure 5C	*Sistotremastrum guttuliferum*	89	0.0001	OTU4
	Litter 41		*Peniophorella pubera*	86	0.0019	OTU17
	Litter 41		*Peniophorella praetermissa*	100	0.0001	OTU20
	Litter 41		*Arrhenia* sp.	90	0.0001	OTU34
	Litter 41		*Rhodonia placenta*	60	0.0095	OTU45
	Litter 41		*Sistotremastrum* sp.	100	0.0001	OTU2710
Aged vs. Young Litter (Years Combined)	Litter Aged	Figure 5C,D	*Ganodermataceae* sp.	88	0.0011	OTU199
	Litter Aged		*Coprinellus* sp.	88	0.0017	OTU902
	Litter Aged		Dacrymyces capitatus	87	0.0081	OTU23
Aged vs. Young Litter (Years Separate)	Litter Aged 41	Figure 5C,D	*Peniophora* sp.	68	0.0096	OTU578
	Litter Aged 25		*Polyporus* sp.	79	0.0072	OTU599
	Litter Aged 25		*Hebeloma* sp.	72	0.0025	OTU724
	Litter Aged 25		*Hyphodermella* sp.	83	0.0096	OTU815
	Litter Aged 25		*Coprinopsis* sp.	77	0.0026	OTU929
	Litter Aged 25		*Lepista* sp.	97	0.0026	OTU1081
Wood 25 vs. 41 (Age Combined)	Wood 41	Figure 5B,C	*Peniophorella praetermissa*	98	0.0001	OTU20
	Wood 41		*Subulicystidium brachysporum*	93	0.0001	OTU14
	Wood 41		*Arrhenia* sp.	73	0.0003	OTU34
	Wood 41		*Rhodonia placenta*	64	0.0014	OTU45
	Wood 25		*Sistotrema* sp.	75	0.0012	OTU43
Wood Samples (Litter Type Separate)	Wood Aged Litter	Figure 5A,C	*Peniophorella pubera*	97	0.0002	OTU17
	Wood Aged Litter		*Dacrymyces capitatus*	72	0.0014	OTU89
	Wood No Litter		*Dacrymyces* sp.	84	0.0031	OTU5
Wood Samples (Litter and Year Separate)	Wood No Litter 25	Figure 5A,C	*Sistotrema* sp.	90	0.0007	OTU43
	Wood Young Litter 41		*Sistotremastrum* sp.	63	0.0051	OTU2710
	Wood Young Litter 41		*Arrhenia* sp.	95	0.0036	OTU34
All Leaf and Wood Treatments Separate	Litter Aged 25	Figure 5C	*Hebeloma* sp.	60	0.0066	OTU724
	Litter Aged 25		*Polyporus* sp.	92	0.0052	OTU599
	Litter Aged 25		*Hyphodontia* sp.	66	0.0053	OTU650
	Litter Aged 25		*Hyphodermella* sp.	91	0.0015	OTU815
	Litter Aged 25		*Coprinopsis* sp.	62	0.0065	OTU929
	Litter Aged 25		*Lycoperdon* sp.	70	0.0067	OTU935
	Litter Aged 25		*Coprinellus* sp.	96	0.0009	OTU902
	Litter Aged 25		*Coprinellus* sp.	83	0.0012	OTU1160
	Litter Aged 25		*Ceriporia* sp.	75	0.0044	OTU997
	Litter Aged 25		*Parmastomyces* sp.	75	0.0048	OTU1105
	Litter Aged 25		*Lepista* sp.	100	0.0002	OTU1081
	Litter Aged 25		*Pluteus* sp.	70	0.0075	OTU1582
	Litter Aged 25		*Crepidotus* sp.	75	0.0067	OTU1612
	Litter Aged 41		*Cylindrobasidium* sp.	75	0.0040	OTU1573
	Litter Aged 41		*Phlebiella* sp.	75	0.0040	OTU1642
	Litter Aged 41		*Trichaptum* sp.	75	0.0040	OTU1794
	Litter Aged 41		*Stereum* sp.	75	0.0040	OTU1520
	Wood Aged Litter 25		*Dacrymyces* sp.	60	0.0081	OTU3403
	Wood No Litter 25		*Sistotrema* sp.	84	0.0004	OTU43
	Wood Young Litter 41		*Sistotremastrum* sp.	89	0.0035	OTU2710

**Table 4 microorganisms-08-00696-t004:** Results of richness analysis. Statistical significance of each richness variable is included in the table, samples with same letter are not significantly different, while those designated with different letters were deemed significant (*p* = 0.00001).

Treatment	Time (mos)	N	S = Richness	E = Evenness	H = Diversity	D = Simpson’s Diversity
Young Control Leaf Litter	NA	2	365.5 BC	0.415 A	2.450 A	0.8381 A
Aged Control Leaf Litter	NA	2	385.5 BC	0.202 A	1.198 A	0.5007 A
Young Litter	25	4	367.0 BC	0.406 A	2.398 A	0.8032 A
Aged Litter	25	4	533.8 AB	0.423 A	2.665 A	0.7350 A
Young Litter	41	4	494.8 AB	0.392 A	2.432 A	0.8076 A
Aged Litter	41	4	719.8 A	0.374 A	2.479 A	0.6990 A
Wood No litter	25	4	130.0 CD	0.592 A	2.875 A	0.8967 A
Wood + Young litter	25	4	123.5 D	0.457 A	2.179 A	0.7859 A
Wood + Aged litter	25	4	135.0 CD	0.422 A	2.069 A	0.7524 A
Wood No litter	41	4	102.8 D	0.452 A	2.102 A	0.7292 A
Wood + Young litter	41	4	100.0 D	0.564 A	2.594 A	0.8498 A
Wood + Aged litter	41	3	86.7 D	0.529 A	2.357 A	0.8192 A

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
