# Peer review of "Role of Leaf Litter in Above-Ground Wood Decay"

_microorganisms, 2020, doi:10.3390/microorganisms8050696_

Round 1

Reviewer 1 Report

Dear Authors,

I appreciated reading your manuscript. The topic is higly relevant and of interest to the journal readers. The manuscript is nicely written and the data clearly discussed. Unfortunately the figures are of low quality. Therefore I suggest revision of your manuscript by including high quality and high resolution figures. Figure 5 is not readable please improve this figure.

Reviewer 2 Report

Dear authors,

The paper discusses an important topic, but requires corrections.
It would certainly be worthwhile to expand/rewrite the discussion. At the moment, it contains some results but not much discussion with the literature. I would also think about the title, because the present seems to be too general. The paper is in many places incorrectly formatted (abstract, tables, references). Figures 5 and 6 also need to be corrected. They are illegible.

Here are my comments.

TITLE

The title seems too general to me. Only one type of tree was studied - southern pine, and only leaf litter, when organic detritus may have different meanings depending on the environment. In the forests it is typically dominated by leaf, but there is also organic matter from other plant and animal remains, twig, bacteria, faeces, etc. In my opinion it is worth to specify the title. Maybe: „Role of organic detritus (leaf litter) in above-ground wood decay” or „Role of leaf litter in above-ground wood decay”.

LIST OF AUTHORS

Please correct. Journal requirement:
Firstname Lastname 1, Firstname Lastname 2 and Firstname Lastname 2,*

ABSTRACT

Abstract is too long. According to the journal instructions, it should have max 200 words, this has 234.

L26-29: I think there is no need to indicate in the abstract that the results obtained are consistent with the literature.

L29-30:  the “detritus” has already been explained at the beginning, there is no need to reinsert the explanation in parentheses.

INTRODUCTION

L53-66: In the first sentence several references (3-7) are given, while the remainder of the paragraph is completely devoid of them. E.g. to L58.

MATERIALS & METHOD

L119-123: Such an introduction seems unnecessary. Details should be given on the methods and general information at the end of the introduction.

L121-123: This information should be placed at2.5. Amplicon based sequences analysis. 

L125-127: References?

L127: „we chose” - Looks bad. Please rewrite this sentence.

L164: space “N= 25”

L191: the measurements were taken in repetitions?

L207 : please add a space between “O = failed” etc.

L247: space “30 sec”

L264: Here is where "OTU" appears for the first time, and nowhere is the abbreviation explained, please add.

RESULTS

Tables (1,2 3, 4)– are not formatted according to the requirements of the journal. Part of the signature should be in the footer under the table, e.g. at table 1 the explanation of symbols a and b should rather be under the table.

Figure 3 – Please sign the X-axis. Please state what was n for the average in your signature. Why are there no error bars? L317 – italics are unnecessary “Figure 3”

Figure 4 – The graph would look better if the Y-axis was lowered, i.e. the scale can start with e.g. 4.0 and thus the lines would be more visible. Please state what was n for the average in your signature. Why are there no error bars? L336 – italics are unnecessary “Figure 4”

L339: Two dots at the end of the sentence

L352-360: different font size

L352: “wood from 25 months (24 mos.)” shouldn't be “wood from 25 months (25 mos.)”?

L360, 397, 368, 369, 370, : please add a space between “p = 0.009” etc.

Figure 5: I don't know what it looks like on the original graphics, but it's completely illegible to me. I don't see any signatures on the charts or legends. At 200% zoom, they're still illegible and blurry.

L404: “IV >/= 60%” – please use the mathematical symbol „≥”

L408-415: It seems to me that formulas should be in the materials and methods section.

Table 4: I would suggest inserting letters with different values after a space or as a superscript or in parentheses, at present they are too much mixed with digits. Please also harmonize the entries - there is one entry with one decimal digit, one with two, and three and four are. Please use a specific entry in a given column. This is then more readable.

L426, L427: In paper is a numerical record everywhere. Please apply here too (“64 OUTs”; “12 OTUs”).

L430: “class” no “Class”

Figure 6: I don't know what it looks like on the original graphics, but it's completely illegible to me. I don't see any signatures on the charts (%) or legends. At 200% zoom, they're still illegible. The abbreviations used in the legend should, in my opinion, be explained under the figure, not in the text itself, but directly in the description of the figure. There are no A-G letters in the diagrams, which the authors refer to under the figure. The letter "F" is used twice, there should still be "G".

DISCUSSION

L469-480: No references to literature, no discussion; just speculation;

L482: References? „many studies” – please indicate any.

L498-513: This fragment is like describing the results. Not like a discussion. There is no comparison of these results with the literature.

L500: some mistake with parentheses - not closed?

L501-502: references?

L504-505: Ganodermataceae sp., Coprinellus sp., Dacrymyces sp.

L507-508 – add comma between fungi name

L508: Hyphodermella spp., Corinopsis spp.

L509: “spp.”

L524-562: That's a very good subchapter. This is a valuable discussion of results.

CONCLUSIONS

L566-569: Perhaps the most important conclusions would be clearer?

L585: “aged litter.” – add dot

REFERENCES

All references are poorly formatted; no bold year, no italics in the magazine title. In many places there are no dots after initials, year of publication placed in the wrong place. I don't know why months are added (there are no months in the template). There are "and" added before the last author.

L637, no. 10: Two into one line/number? It should be: 10. Hammel K.E. (1997) and 11. Abarenkov K. (2010) /

  1. 49 - Caps lock?
  2. 53 – date of completion?

The correct format is as follows:

  1. Author 1, A.B.; Author 2, C.D. Title of the article. Abbreviated Journal Name Year, Volume, page range.
  2. Author 1, A.; Author 2, B. Title of the chapter. In Book Title, 2nd ed.; Editor 1, A., Editor 2, B., Eds.; Publisher: Publisher Location, Country, 2007; Volume 3, pp. 154–196.
  3. Author 1, A.; Author 2, B. Book Title, 3rd ed.; Publisher: Publisher Location, Country, 2008; pp. 154–196.
  4. Author 1, A.B. Title of Thesis. Level of Thesis, Degree-Granting University, Location of University, Date of Completion.

L725, no. 44: no newer paper on the subject? The citation is 47 years old, it is worth looking for something much newer.  Especially since the two references used in the sentence L465-466 are older, i.e. from 1973 (44) and 1992 (43). And the topic of wood moisture is important and it is worthwhile to touch on newer researches in this field.

Interesting publications  related to the topic:

Schmidt O.: Wood and tree fungi; Springer-Verlag Berlin Heidelberg, 2006; https://doi.org/10.1007/3-540-32139-X

Stienen T., Schmidt O., Huckfeldt T.: Wood decay by indoor basidiomycetes at different moisture and temperature. Holzforschung, 2014, 68(1):9-15; 10.1515/hf-2013-0065

Goodell B., Qian Y., Jellison J.: Fungal Decay of Wood: Soft Rot—Brown Rot—White Rot. ACS Symposium Series, 2008, 982: 9-31; 10.1021/bk-2008-0982.ch002

Rajala T., Tuomivirta T., Pennanen T., Makipaa R.: Habitat models of wood-inhabiting fungi along a decay gradient of Norway spruce logs. Fungal Ecology, 2015, 18:48-55; https://doi.org/10.1016/j.funeco.2015.08.007

Ringman R., Beck G., Pilgard A.: The Importance of Moisture for Brown Rot Degradation of Modified Wood: A Critical Discussion. Forests, 2019, 10(6), 522; https://doi.org/10.3390/f10060522

Zelinka S.L., Kirker G.T., Bishell B., Glass S.V.: Effects ofWood Moisture Content and the Level of Acetylation on Brown Rot Decay. Forests, 2020, 11,299; doi:10.3390/f11030299
